# T Helper Cells: The Modulators of Inflammation in Multiple Sclerosis

**DOI:** 10.3390/cells9020482

**Published:** 2020-02-19

**Authors:** Martina Kunkl, Simone Frascolla, Carola Amormino, Elisabetta Volpe, Loretta Tuosto

**Affiliations:** 1Department of Biology and Biotechnology Charles Darwin, Sapienza University, 00185 Rome, Italy; martina.kunkl@uniroma1.it (M.K.); frascolla.1688916@studenti.uniroma1.it (S.F.); amormino.1696927@studenti.uniroma1.it (C.A.); 2Laboratory affiliated to Istituto Pasteur Italia-Fondazione Cenci Bolognetti, Sapienza University, 00185 Rome, Italy; 3Neuroimmunology Unit, IRCCS Santa Lucia Foundation, 00143 Rome, Italy; e.volpe@hsantalucia.it

**Keywords:** multiple sclerosis, inflammation, T helper cells, immunotherapy

## Abstract

Multiple sclerosis (MS) is a chronic neurodegenerative disease characterized by the progressive loss of axonal myelin in several areas of the central nervous system (CNS) that is responsible for clinical symptoms such as muscle spasms, optic neuritis, and paralysis. The progress made in more than one decade of research in animal models of MS for clarifying the pathophysiology of MS disease validated the concept that MS is an autoimmune inflammatory disorder caused by the recruitment in the CNS of self-reactive lymphocytes, mainly CD4^+^ T cells. Indeed, high levels of T helper (Th) cells and related cytokines and chemokines have been found in CNS lesions and in cerebrospinal fluid (CSF) of MS patients, thus contributing to the breakdown of the blood–brain barrier (BBB), the activation of resident astrocytes and microglia, and finally the outcome of neuroinflammation. To date, several types of Th cells have been discovered and designated according to the secreted lineage-defining cytokines. Interestingly, Th1, Th17, Th1-like Th17, Th9, and Th22 have been associated with MS. In this review, we discuss the role and interplay of different Th cell subpopulations and their lineage-defining cytokines in modulating the inflammatory responses in MS and the approved as well as the novel therapeutic approaches targeting T lymphocytes in the treatment of the disease.

## 1. Introduction

Multiple sclerosis (MS) is a chronic inflammatory autoimmune disorder of the central nervous system (CNS) affecting about 2–3 million people worldwide that is triggered by both environmental and genetic factors [1,2]. About 15–30% of patients with MS present the relapsing-remitting (RR) clinical course, which is characterized by acute episodes of neurological dysfunctions, such as optic neuritis, sensory disturbances, or motor impairments, usually followed by periods of recovery or remission [3]. After variable periods of time, about 50% of RRMS patients progress to a chronic secondary progressive (SP) clinical stage that is characterized by steadily worsening disability [4]. In about 15% of patients, MS is progressive from the onset and is called primary progressive (PP)MS, a clinical course characterized by a gradual and constant decline in neurological functions [5].

The pathological hallmarks of MS are the breakdown of the blood–brain barrier (BBB), oligodendrocyte loss, demyelination, astrocytes gliosis, and axonal degeneration [6,7]. Inflammation is present at all stages, and pro-inflammatory cytokines and chemokines play a critical role in the pathophysiology of MS by compromising the BBB, recruiting immune cells from the periphery and activating resident microglia. Microglia activation is thought one of the early events in the development of MS lesions. Activated microglia, indeed, may further contribute to disease progression by secreting inflammatory cytokines and chemokines and by releasing reactive oxygen species and glutamate [8]. Conversion of MS from RR to the progressive phase has also been related to prolonged chronic inflammation in the CNS. Moreover, both SPMS and PPMS patients have generalized inflammation in the whole brain accompanied by cortical demyelination and diffuse white matter injury [9]. Although every cell type of the innate and adaptive immune system may orchestrate the inflammatory response within the CNS, a significant and important contribution is exerted by autoreactive CD4^+^ T cells. Autoreactive CD4^+^ T cells likely activated in the peripheral lymph nodes migrate into the CNS [10,11,12,13,14] where they are locally reactivated and secrete cytokines and chemokines that modulate the inflammatory lesions typical of MS [15]. For instance, the strongest genetic risk factor for MS is human leucocyte antigen (HLA)-DRB*15:01, a major histocompatibility complex (MHC) class II allele involved in the presentation of self-peptides to CD4^+^ T cells [16]. The aim of this review is to provide a detailed and comprehensive description of the role of different CD4^+^ T helper (Th) cell subsets in the pathophysiology of MS and the current therapeutic approaches targeting T-cell mediated responses. The role of regulatory T (Treg) cells in suppressing the functions of autoreactive Th cells in MS is also briefly discussed.

## 2. Th Cell Subsets

CD4^+^ Th cells are central regulators of the adaptive immune response against a wide variety of microbes by helping B lymphocytes to produce antibodies (Ab) and by secreting specific cytokines that provide efficient protection against pathogens. Distinct Th cell subsets, producing one or more lineage-defining cytokines and expressing master transcription factors and homing receptors, differentiate from naïve CD4^+^ T cells in response to a specific class of pathogenic microorganisms and to the cytokine milieu. Naïve CD4^+^ T cells are activated in peripheral lymph nodes by mature dendritic cells that present pathogen-derived peptides associated to class II major histocompatibility complex (MHC) and together with costimulatory molecules promote T cell proliferation and produce polarizing cytokines, which in turn orchestrate T cell differentiation in distinct Th cell subsets, such as Th1, Th2, Th17, Th22, and Th9 [17,18]. In addition to their protective role against pathogens, specific Th cell subsets exert a crucial role in MS pathogenesis as detailed below.

### 2.1. Th1 Cells

Th1 cells were identified in the late 1980s [19,20] as a subset of CD4^+^ T cells that orchestrate efficient adaptive immune responses against intracellular pathogens by secreting interferon (IFN)-γ that activates macrophages to kill intracellular microbes and promotes the production of opsonizing Abs [17]. Th1 may be identified by the surface expression of the CXC chemokine receptor type 3 (CXCR3), and interleukin (IL)-12 receptor (IL-12R) chains β1/β2, together with the intracytoplasmic expression of the master transcription factor T-bet [21]. IL-12 and IFN-γ cooperate for inducing the expression of T-bet that in turn induce additional IFN-γ and T-bet expression, thus amplifying Th1 polarization [22,23]. Initial studies performed in an experimental autoimmune encephalomyelitis (EAE) animal model of MS revealed a pivotal role of Th1 cells in the pathogenesis of MS. Indeed, IFN-γ-producing Th1 cells were found as the most frequent Th cell subset in the CNS of EAE animals [24,25]. IFN-γ was abundant in CNS lesions of EAE [26] and in active lesions of MS patients [27], and the adoptive transfer of Th1 cells was sufficient to induce EAE in recipient mice [28,29]. These data together with the initial findings that the administration of IFN-γ to MS patients exacerbated disease [30,31] supported an important role of IL-12/IFN-γ axis and Th1 cells in both EAE and MS pathogenesis. Although the neuropathological role of infiltrated Th1 cells into the CNS is not fully elucidated, several evidences suggest microglia as the major cellular targets of Th1 cells. Microglia are CNS resident macrophages that, depending on the nature of exogenous stimuli, may differentiate into inflammatory M1-like or anti-inflammatory M2-like phenotypes [32]. Th1 cells produce several effector molecules that activate resident microglia and trigger their differentiation into the inflammatory and neurotoxic M1-like phenotype [33,34]. Moreover, Th1 cells also favor the upregulation of class II MHC and costimulatory molecules on microglia, thus favoring the reactivation of infiltrated Th cells and their further differentiation [34].

However, further observations that mice deficient in IL-12p35 subunit [35,36] or IL-12Rβ2 chain [37] or IFN-γ [38] were susceptible to EAE together with data showing that the administration of IL-12 during the early phases the disease suppressed EAE in an IFN-γ-dependent manner [35], undermined the paradigm that Th1 cells were the main pathogenic cell subset in EAE and MS. Finally, the discovery that IL-23 shares the p40 subunit with IL-12 [39] and IL-23R comprises the IL-12Rβ1 chain [40] clarified these contradictory data and defined the role of IL-23 in favoring EAE by driving and inducing the expansion of Th17 cells, a subset of Th cells producing IL-17 [41,42,43].

### 2.2. Th17 Cells

Th17 cells have been identified in 2005 [41] as an important subset of CD4^+^ Th cells that mediates efficient immune responses against extracellular bacteria and fungi [44]. Th17 cells may be identified on the basis of specific surface markers such as CD161, the chemokine receptors CCR6 and CCR4, the cytokine receptors IL-23R and IL-1R [45], the intracytoplasmic expression of retinoic acid receptor-related orphan nuclear receptor γτ (RORγτ), and the production of specific cytokines, such as IL-17A-F, IL-21, and IL-22 [46]. In particular, IL-17A and IL-17F act on several cell types like macrophages, epithelial, and endothelial cells by inducing the expression of inflammatory cytokines and chemokines and promoting the recruitment of neutrophils in inflammatory sites [47,48]. IL-21 provides a positive-feedback loop for successful amplification of the Th17 cell subset [49], and IL-22 acts on epithelial cells favoring the production of antimicrobial peptides and mucus [50]. The result of these actions is the promotion of host defense against pathogens but also tissue damage that may lead to chronic inflammatory diseases [51]. Notably, Th17 cells are involved in the pathogenesis of several autoimmune diseases including MS [52], where they critically contribute to the disruption of the BBB [53] and by targeting resident astrocytes and microglia within the CNS promote their activation and amplify neuroinflammation in EAE [33,54] (Figure 1). In particular, Th17 cells are highly effective in regulating astrocytes rather than microglia. Astrocytes are CNS resident cells with distinct anatomical locations and both morphological and functional properties. They are positioned at the interface between the BBB and neurons and regulate the movement of molecules and cells between circulation and CNS. Moreover, by producing neurotrophic factors they also regulate neurogenesis and tissue repair. In response to CNS injury, astrocytes undergo profound morphological and functional changes, a phenomenon known as astrogliosis [55]. Astrocytes express a functional IL-17 receptor A that is further upregulated in EAE [56]. IL-17 has been described to upregulate the production of inflammatory cytokines and chemokines [57], and the impairment of IL-17-mediated signaling in astrocytes has been described to ameliorate EAE [58]. Consistently, Th17 cells, in cooperation with Th1, have been shown to regulate the functions of astrocytes by inducing the downregulation of neurotrophic factors and the upregulation of inflammatory cytokines and chemokines [54,59]. Other proposed pathological functions of Th17 cells and IL-17 include the inhibition of both maturation and survival of oligodendrocytes (OLs) [60] as well as their apoptosis [61]. OLs are myelinating glial cells of the CNS during development and throughout adulthood. In MS, persistent demyelination and neurodegeneration are associated with dysfunction and apoptosis of OLs caused by direct cytotoxicity from antigen-specific T cells and autoantibodies as well as by T cell-mediated pro-inflammatory cytokines that activate resident microglia [62].

During the past years, several studies in EAE mice [15,41,63] together with the immunological evaluation of MS patients evidenced a pivotal role of Th17 cells in the pathogenesis of MS [42,43,64,65]. High levels of IL-17-producing CD4^+^ T cells have been found in the peripheral blood or cerebrospinal fluid (CSF) of RRMS patients during relapses [65,66,67] and in MS patients with active disease [68]. Moreover, the preferential expansion of Th17 cells has been correlated with the number of active plaques on magnetic resonance imaging (MRI) [69] and with disease progression [70]. IL-17A in combination with IL-6 has been also described to promote the breakdown of BBB in RRMS patients [71] by altering the expression of adhesion molecules on endothelial cells [72,73] and favoring the depolymerization of the actin cytoskeleton near the tight junctions [74]. For instance, human Th17 cells exhibit an increased migratory capacity in RRMS patients with high disease severity and inflammatory lesions [75].

Several key transcription factors, the most reliable RORγτ [76] and STAT3 [77,78], and cytokines such as IL-6, transforming growth factor (TGF)β, IL-21, IL-1β and IL-23, have been described to promote IL-17 expression and Th17 cell differentiation in both human and mouse [18,45,79,80,81]. In the mouse system, the differentiation of Th17 cells is promoted by TGFβ in combination with either IL-6 or IL-21 or IL-1β [82,83]. In the human system, some research groups suggested that IL-1β and IL-23 can supply for TGFβ functions in inducing human Th17 cell differentiation [84,85,86,87]. Others evidenced a pivotal role of TGFβ in combination with IL-6, IL-1β IL-23, or IL-21 for inducing the expression of RORγτ and Th17 cell differentiation [88,89,90]. In human, but not in mouse [91], CD28, an important co-stimulatory molecule expressed on T cells [92,93] and involved in the regulation of both tolerance and autoimmunity [94], has been also described to stimulate CD4^+^ T cells to produce inflammatory cytokines and chemokines related to the Th17 cell phenotype [95,96] and to enhance the inflammatory response in MS by reprogramming T cell metabolism and favoring the expression of Th17-associated cytokines [97,98].

Th17 cells exhibit high plasticity and may adopt different functional profiles depending on the inflammatory or anti-inflammatory environments [99]. Th17 cells can differentiate into high pathogenic Th1-like Th17 cells expressing high levels of IFN-γ (see below) or in non-pathogenic IL-10-producing Treg cells. Both Th17 and Treg cells, indeed, share some common features, such as the expression of CD49b and the transcription factor aryl hydrocarbon receptor (AhR), likely due to the common TGF-β requirement for their differentiation [100,101]. In this contest, Gagliani et al. showed in mice that, during the resolution of inflammation, Th17 cells undergo a transcriptional reprogramming leading to their differentiation into Treg cells in the presence of TGF-β and in an AhR-dependent manner [102]. The high plasticity of Th17 cells was also recently evidenced in MS by Capone et al., who showed that human Th17 cells polarized from the naïve CD4^+^ T cells as well as from peripheral blood Th17 cells of RRMS patients display distinct inflammatory features compared to healthy donors. In particular, Th17 cells from RRMS patients upregulate the expression of IL-1R and produce higher levels of IL-21, IL-2, and TNF-β thus leading to the acquisition of a more pathogenic profile [103]. Consistently, elevated expression of IFN-γ and CXCR3 together with reduced expression of anti-inflammatory IL-10 was found in Th17 cells from clinically active compared to stable MS patients [104]. Th17 cells also produce granulocyte-macrophage colony-stimulating factor (GM-CSF), another cytokine that has been suggested to play a critical role in the pathogenicity of Th17 cells in EAE models [52]. Moreover, the percentage of IL-17A and GM-CSF co-producing cells were enriched in the CSF of RRMS [105] and increased during relapse [106]. Th17 cells have been also shown to produce IL-22 [50], a cytokine that raises pro-inflammatory innate defense mechanisms in epithelial cells [107,108] and that, together with other cytokines of the Th17 signature, has been involved in several inflammatory and autoimmune diseases [109,110].

### 2.3. Th1-Like Th17 Cells

Recently, a novel Th1 cell subset known as Th1-like Th17 cells that produce both IFN-γ and IL-17 has been identified in both human and mice [111,112]. These cells express IL-23R and co-express CXCR3 and T-bet together with CCR6 and RORγτ, produce lower amounts of IL-17A compared to classical Th17 cells but high levels of IFN-γ [45]. The origin of Th1-like Th17 cells is not fully defined but several studies support the notion that these cells derive from Th17 cells in the presence of IL-12, TNF-α and/or, IL-1β [113,114,115]. In mice, the higher pathogenicity of Th1-like Th17 cells compared to Th17 cells was associated with the production of several inflammatory cytokines and chemokines such as GM-CSF, IL-22, CC chemokine ligand 4 (CCL4), and CXCR3 [116]. These data were also confirmed in the human system, by the analysis of the transcriptional profile of peripheral blood Th17 cells and Th1-like Th17 cells in MS [104]. Recent data from single-cell transcriptome analysis performed in EAE extended these data by showing that the differentiation of Th1-like Th17 cells in the CNS from Th17 precursors in the lymph nodes was governed by four specific genes, Gpr65, Toso, Plzp, and Cd5l, which were also involved in disease susceptibility [117].

The participation of highly pathogenic Th1-like Th17 cells in neuroinflammation has been evidenced by studies in both EAE and MS [118]. For instance, Th1-like Th17 cells were capable to cross the BBB and accumulate in the CNS of acute EAE and were also found in brain tissues from MS patients [119] and upregulated in RRMS patients during relapse [120]. Myelin-specific Th1-like Th17 cells increased in both peripheral blood and CSF of patients with MS [121,122] and Th17.1 cells, a Th1-like Th17 subpopulation expressing high levels of IFN-γ, GM-CSF, very late antigen 4 (VLA-4), and low levels of IL-17, have been associated with the disease activity in RRMS patients [123]. The neuropathic effects of Th1-like Th17 cells within CNS are still being elucidated but likely overlap with those identified for Th1 and Th17 cells (Figure 1).

### 2.4. Th22 Cells

Discovered in 2000 [124], IL-22 is produced by Th17 cells and by a unique CD4^+^ Th cell subset, named Th22 [125,126,127]. Other cellular sources of IL-22 include natural killer (NK) cells [128,129], ILC3 subset of innate lymphoid cells, and, at a lower extent, other leucocytes including macrophages [130]. IL-22 is a member of the IL-10 family and its main effects are exerted in non-hematopoietic cells, such as epithelial cells. IL-22, indeed, stimulates the regeneration and proliferation of epithelial cells and promotes the production of antimicrobial peptide required for epithelial barrier functions and protection against extracellular pathogens [131].

Although the role of IL-22 in MS has not yet been fully elucidated, emerging evidence suggest the involvement of IL-22 in MS immunopathogenesis. Initial studies from Kebir et al. evidenced the upregulation of IL-22R in the brains of MS patients and the role of IL-22 in synergy with IL-17A in disrupting the integrity of BBB tight junctions by reducing the expression of occludin in endothelial cells [71]. More recent data showed higher levels of IL-22 in the serum of relapsing MS compared to healthy donors [106,132,133,134] and a decrease of IL-22 was observed during the recovery phase of acute EAE [135]. The number of IL-22-producing cells also increased in both peripheral blood [136] and CSF [105] of RRMS patients who were resistant to IFN-β therapy [136]. Due to high levels of expression of CCR6, which is required for the migration into the CNS, myelin-specific IL-22-producing cells can synergize with Th17 cells, thus contributing to break the BBB and initiate the autoimmune response against components of the CNS myelin [71]. For instance, myelin-specific IL-17- and IL-22-producing CD4^+^ T cells resistant to corticoids have been associated with active brain lesions in MS patients [134]. Moreover, IL-22R expression was found in human brains and co-localized with IL-22 in the brain tissue sections of MS patients, in particular in the plaques and predominantly on astrocytes [133], which may increase the permeability of the BBB by secreting matrix metalloproteinases (MMPs) [55]. Interestingly, recent studies suggest a role of IL-22 and IL-22-producing cells in regulating the survival and activity of both OLs and astrocytes in MS [133,137] (Figure 1). Finally, the identification of single nucleotide polymorphism of IL-22 binding protein (IL-22BP, also called IL-2RA), an antagonist of IL-22, as an MS risk gene [138,139] supports a role of IL-22-producing cells in the immunopathogenesis of MS.

### 2.5. Th9 Cells

Over the past decade, a distinct subset of effector CD4^+^ T cells characterized by the production of IL-9 has been defined. Although IL-9 was initially associated with a Th2 response, more recent studies redefined IL-9-producing CD4^+^ T cells as Th9 cells [140,141]. The role of Th9 cells in MS has been largely investigated in the mouse model of EAE. The first study compared the encephalitogenic activity of myelin oligodendrocyte glycoprotein (MOG)-specific Th9, Th17, and Th1 cells in an EAE model of adoptive transfer. The results revealed that Th9 cell recipients possessed fewer infiltrates of lymphocytes in the meninges compared to EAE developed by Th1 and Th17 cells, thus suggesting that the mechanisms of EAE induction by Th9 cells differed from the mechanisms of Th1 and Th17 cells [142]. Further studies performed in knockout mice reported contrasting results. In fact, mice lacking IL-9R exhibited a more severe course of EAE, suggesting a regulatory role of the cytokine in controlling pathogenic mechanisms of immune responses [143]. Consistently, in EAE, IL-9 was reported as an anti-inflammatory cytokine produced by nonpathogenic Th17 cells concomitantly to IL-10 [144]. On the other side, other groups found that IL-9 neutralization and IL-9R deficiency attenuated EAE [145,146].

In the human system, Ruocco et al. analyzed the effect of IL-9 in MS by correlating the levels of IL-9 in the CSF of 107 RRMS patients at the diagnosis and during the course of disease. Interestingly, they found that IL-9 levels in the CSF of RRMS patients inversely correlated with indexes of inflammatory activity, neurodegeneration, and progression of MS-associated disability [147]. A previous study in MS patients showed that IL-9 levels in the CSF were lower during clinical relapses and increased following prednisolone treatment, thus supporting the role of IL-9 in the maintenance of the remission phase in MS [148]. These results suggest the presence of Th9 cells in the CNS of MS patients, where IL-9 could exert a protective role in MS disease by reducing the levels of IL-17 produced by Th17 cells [147] and the expression of the mitochondrial pro-apoptotic factor Bax [149], thus enhancing neuronal survival in organotypic human brain slice cultures [150]. Considering the contrasting results reported on the contribution of Th9 cells and IL-9 to either MS or EAE, further studies are required to clarify the role of Th9 cells in MS pathogenesis.

## 3. Treg Cells

The development of autoimmunity is finely regulated by specific subsets of T cells that modulate immune responses by keeping pathogenic Th cells under control. Treg cells were firstly identified in 1995 by Sakaguchi et al. as a CD4^+^CD25^+^ T cell subset able to suppress effector inflammatory T cells and to maintain self-tolerance [151]. In the last two decades, progress has been made to better characterize Treg cells in the context of autoimmune diseases and at least two main subsets of CD4^+^ Treg cells have been identified: natural nTreg cells that develop in the thymus following the recognition of self-antigens and express the transcription factor forkhead box P3 (FoxP3) [152] and inducible iTreg cells that generate from naïve CD4^+^ T cells under specific conditions of antigen-stimulation and in the presence of a particular cytokine milieu. iTreg comprises different subsets including FoxP3^+^ Treg, type 1 regulatory (Tr1) T cells producing high levels of IL-10 and Th3 cells producing TGF-β [153]. In humans, circulating iTreg also exhibit high levels of phenotypic heterogeneity strictly related to their differentiation stage and immunosuppressive functions [154].

Alterations in both number and suppressive functions of distinct Treg subsets have been reported in MS, especially in RRMS patients [155,156], and most of the approved disease-modifying drugs for RRMS also exert positive effects on Treg cells [154].

## 4. T Cell Targeting Therapy

The relevant contribution of autoreactive inflammatory Th cell subsets in the pathogenesis of MS is supported by the fact that most of the approved disease-modifying therapy target T cells.

IFN-β an antiviral cytokine with immunosuppressive and anti-proliferative effects, has been the first-line drug approved by the Food and Drug Administration (FDA) for the treatment of RRMS in 1993. The most relevant effects of IFN-β on T cells are the inhibition of T cell activation and secretion of inflammatory cytokines [157], the shift of Th cells from an inflammatory Th1 to an anti-inflammatory Th2 cell phenotype, the reduction of both Th17 and IL-17 levels, and an increase of suppressive Treg [158,159]. A significant decrease of IL-22 in the serum of RRMS patients after 6 and 12 months of treatment with IFN-β has been also recently reported and correlated with a decrease of MS severity [160].

Glatiramer acetate (GA), a mixture of synthetic polypeptides containing only 4 amino acids, glutamate, lysine, alanine, and tyrosine, was approved by the FDA for the treatment of RRMS in 1996. Although the mechanisms of action of GA are not completely understood and elucidated, several studies suggest a pivotal role of GA on different antigen-presenting cells by promoting the differentiation of microglia, macrophages, and dendritic cells into the suppressive M2 phenotype producing anti-inflammatory cytokines, such as TGF-β and IL-10 [161,162,163]. The results of these actions on T cells involve the shift of reactive T cells from Th1 to Th2 [164,165] and the increase of Treg [166] that suppress the activity of all autoreactive inflammatory Th cell subsets. In this context, recent data from Spadaro et al. demonstrated that in GA-responder RRMS patients, anti-inflammatory Treg cells persist for more than 10 years after GA treatment [167]. Similar to IFN-β GA is not effective in reducing the disability in progressive MS [168,169] but, for its long-term efficacy and safety, is the most common first-line therapy registered worldwide for RRMS [170].

Dimethyl fumarate (DMF), a fumaric acid ester with anti-inflammatory activity has been approved for the treatment of RRMS in 2013 [171]. DMF exerts both direct and indirect effects on T cells [172]. DMF reduces the production of inflammatory cytokines, such as IL-1β, TNF, and IL-6 in microglia [173] and inhibits inflammatory cytokine production in human peripheral blood mononuclear cells [174,175], probably by interfering with the activation of nuclear factor (NF)-κB family of transcription factors [176]. In addition to a general reduction in peripheral T cell numbers observed in RRMS patients treated with DMF [177,178], profound changes in circulating Th cell subsets were also observed [179] with a particular reduction of memory CD4^+^ T cells [178]. DMF treatment also induces a shift of the balance between different Th cell subsets, with a reduction of IFN-γ-producing Th1 and IL-17-producing Th17 cells and an increase of IL-4-producing Th2 cells [180]. Finally, an optimal response to DMF treatment was also associated with an increase of Treg cells in the peripheral blood of RRMS patients [179,181].

Fingolimod, a first orally therapeutic molecule approved for RRMS [182,183], acts as an antagonist of sphingosine-1-phosphate receptor (S1PR) that is essential for the egress of T lymphocytes from lymph nodes [184,185]. By binding S1PR and inducing its internalization, fingolimod mediates the sequestration of both naïve and effector memory T lymphocytes in the lymphoid organs [186,187]. A detailed analysis of fingolimod efficacy on different T cell subsets in treated RRMS patients evidenced a selective reduction in the frequency of both IFN-γ- and IL-17-producing cells [188]. More recent data from Dominuguez-Villar et al. showed that fingolimod treatment strongly reduced central memory Th1, Th17, and Th1-like Th17 cells, whereas effector memory Th1 and Th1-like Th17 cell subsets were less affected. The analysis of fingolimod effects on T cells during 12 months of treatment also evidenced a modulation of the phenotype of Th cells and Treg cells. Effector Th cells showed a decrease of IL-17 and IFN-γ and an increase of TGF-β and IL-10 production together with the expression of exhaustion markers such as programmed death 1 (PD-1) and T-cell immunoglobulin and mucin-domain containing-3 (TIM3) [189]. Moreover, the suppressive functions of Treg in fingolimod-treated RRMS patients were also upregulated, thus evidencing an additional role of fingolimod in restoring peripheral tolerance besides the retention of T lymphocytes in lymph nodes [189].

Natalizumab is another disease-modifying drug that interferes with the migration of T cells approved for RRMS in 2004. It is a monoclonal antibody (mAb) that targets VLA-4 expressed by activated T and B cells [158] and inhibits VLA-4 interaction with vascular cell adhesion molecule-1 (VCAM-1) expressed on endothelial cells [190], thus blocking T cell penetration of the BBB [191,192,193,194], in particular the Th1-like Th17.1 subset [123]. Despite the efficacy of both fingolimod and natalizumab in peripheral sequestering of inflammatory T cells and preventing their migration across the BBB, respectively, abnormal inflammatory activities and recurrence of clinical relapses have been observed after cessation of both fingolimod [195,196] and natalizumab therapy in MS patients [197,198]. Moreover, prolonged treatment with natalizumab has been also associated with the reactivation of latent John Cunningham virus, thus favoring progressive multifocal leukoencephalopathy [199,200].

FDA and EMA approved two humanized mAbs targeting T cells for RRMS. Alemtuzumab, approved in 2013, targets CD52 that is expressed at high levels on B and T cells [201], thus leading to the depletion of both CD4^+^ and CD8^+^ T cells by Ab-dependent (ADCC) and complement-dependent cytotoxicity (CDC) [158]. Daclizumab, approved in 2016, is a mAb targeting CD25, the high-affinity subunit of IL-2 receptor (IL-2R), expressed on activated T cells and Treg [202]. In contrast to alemtuzumab, daclizumab does not mediate T cell depletion by ADCC or CDC but, by neutralizing IL-2 binding to CD25 and high-affinity IL-2R, inhibits the proliferation of activated T cells without affecting cells that express the low-affinity IL-2R, such as natural killer (NK) cells or resting T cells [203]. Due to the high risk of serious and potentially fatal immune reactions, including hepatitis and inflammatory brain disorders such as encephalitis and meningoencephalitis, in May 2018, the marketing authorization for daclizumab (Zynbrita) was revoked by EMA [204]. In November 2019, EMA also recommended a restrictive use of alemtuzumab (Lemtrada) for RRMS patients with highly active disease despite treatment with at least one disease-modifying therapy [205]. Interestingly, Gingele et al. recently reported that ocrelizumab, a humanized mAb targeting CD20 approved in 2017 for relapsing MS and PPMS and thought to specifically target B cells [206,207], was also effective in depleting a highly activated CD20^+^ T cell subset producing inflammatory cytokines [208]. Since CD20^+^ inflammatory T cells are present in the peripheral blood and brain of MS patients [208,209], their depletion may account for the efficacy of ocralizumab therapy.

In addition to the currently approved T cell targeting therapies, several therapeutic approaches aimed at inhibiting autoreactive inflammatory T cells are currently tested in clinical trials for MS [210]. Among them, the most promising therapies include mAb targeting Th17-related cytokines, in particular IL-17A, which are currently in ongoing trials for MS. In particular, Secukinumab, a fully humanized anti-IL17A mAb aimed at suppressing pathogenic Th17 and Th1-like Th17 cell subset, has been approved in 2005 as first-line therapy for psoriasis [211,212]. The significant reduction (67%) of new magnetic resonance imaging (MRI) lesions observed in 73 treatment-naïve RRMS patients after 24 weeks of treatment with Secukinumab [213] led to a placebo-controlled randomized phase II clinical trial (NCT01874340). However, the trial was terminated by the sponsor based on the development of another anti-IL17A mAb, Ikekizumab, with better activity than Secukinumab for psoriasis [214,215] and superior potential in treating MS. Clinical trials are ongoing.

## 5. Conclusions

In the last two decades, our knowledge of the immunopathogenic mechanisms of MS has strongly improved, thus evidencing a fundamental role of encephalitogenic Th cells, in particular, Th1-Th17 like, Th17, Th22, and GM-CSF-producing CD4^+^ T cells, in initiating and perpetuating the inflammatory responses and the consequent neurodegeneration in MS. Accordingly, most of the approved therapeutic strategies are aimed to attenuate inflammatory T helper cells and emerging more selective T-cell-based therapeutic approaches are currently ongoing in clinical trials for MS.

## Figures and Tables

**Figure 1 cells-09-00482-f001:**
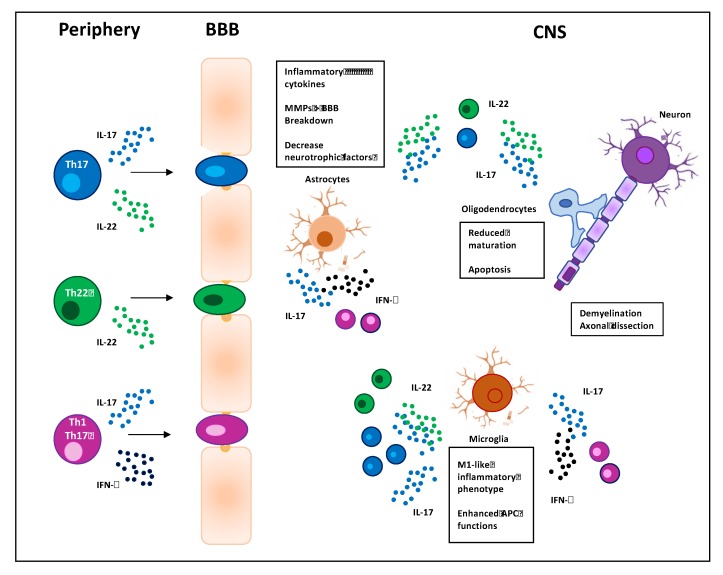
Pathogenic T helper (Th) cell subsets in multiple sclerosis (MS). Self-reactive Th1, Th22 cell, and Th1-like Th17 subsets activated in peripheral lymph nodes cross the blood–brain barrier (BBB) and migrate into the central nervous system (CNS). In the CNS, T cells are reactivated and, by producing their lineage-defining cytokines, regulate the functions of CNS-resident cells (microglia, astrocytes, oligodendrocytes) by enhancing inflammatory cytokine production, antigen-presenting cell (APC) functions, and apoptosis, thus contributing to axonal damage and demyelination.

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
