# Peer review of "T Helper Cells: The Modulators of Inflammation in Multiple Sclerosis"

_cells, 2020, doi:10.3390/cells9020482_

Round 1
Reviewer 1 Report
Kunkl et al outline the contributions of various Th subtypes (Th1, Th17, Th1-like Th17, Th22, Th9) to MS and EAE. They also discuss the impact of various MS therapies on T cell function. The review is topical and of interest; however, there are important gaps in what is discussed. These need to be filled before the manuscript can be considered suitable for publication. Specific comments are below.
I think there ought to be a paragraph break after ref. 9 on line 54 Before jumping in to discuss the various Th subtypes (sec 2), it would be useful to take a step back and explain the operating paradigm in which naïve T cells differentiate into specific subsets in the context of their antigen-specific activation in peripheral lymphoid tissues, and that this differentiation is driven by the cytokine milieu created by APCs, etc. While as immunologists, this may seem basic knowledge, I think that stating this would aid the understanding of those readers who may not be T cell biologists. This paragraph could go right before sec 2, or even as an introductory paragraph to sec 2. Th1: need to mention that T-bet is the master transcription factor for these responses Th1-like Th17 cells: I think this discussion more appropriately belongs after discussing Th17 cells. On the balance, the literature is much more compelling that these cells are originally Th17 origin rather than Th1 Also re; Th1-like Th17; this really needs to be fleshed out and not just the single sentence that precedes ref 41. There is plenty of work suggesting genes and pathways that drive pathogenicity of these cells, and these need to be discussed. See work of Aviv Regev (Gaublomme Cell 2015) among others Re: Th17: what about the reciprocity between Th17 and Treg development (ie role of TGFb, AHR)? This needs to be discussed Line 173, seems better to say “fully” elucidated as the authors go on to discuss evidence for IL-22 involvement in MS Re therapies: daclizumab was pulled from the market worldwide in 2018!Author Response
We thank the reviewer for suggestions and advices. To better clarify our responses, we numbered the reviewer’s comments and our responses are indicated below
1. I think there ought to be a paragraph break after ref. 9 on line 54. Before jumping in to discuss the various Th subtypes (sec 2), it would be useful to take a step back and explain the operating paradigm in which naïve T cells differentiate into specific subsets in the context of their antigen-specific activation in peripheral lymphoid tissues, and that this differentiation is driven by the cytokine milieu created by APCs, etc. While as immunologists, this may seem basic knowledge, I think that stating this would aid the understanding of those readers who may not be T cell biologists. This paragraph could go right before sec 2, or even as an introductory paragraph to sec
Response: We are grateful to the reviewer for the suggestion. In the revised manuscript, we added a paragraph before describing the different Th cell subsets where we briefly described how naïve CD4+ T cells are activated in lymph nodes and differentiate into distinct Th cell subsets (new paragraph 2)
2. Th1: need to mention that T-bet is the master transcription factor for these responses
Response: As suggested, we better clarify that T-bet is the master transcription factor for Th1 cells (new paragraph 2.1, lines 83-85)
3. Th1-like Th17 cells: I think this discussion more appropriately belongs after discussing Th17 cells. On the balance, the literature is much more compelling that these cells are originally Th17 origin rather than Th1. Also re; Th1-like Th17; this really needs to be fleshed out and not just the single sentence that precedes ref 41. There is plenty of work suggesting genes and pathways that drive pathogenicity of these cells, and these need to be discussed. See work of Aviv Regev (Gaublomme Cell 2015) among others
Response: As suggested, we better discussed Th1-like Th17 cells by describing genes and pathways driving their pathogenicity (new paragraph 2.3, lines 202-211)
4. Th17: what about the reciprocity between Th17 and Treg development (ie role of TGFb, AHR)? This needs to be discussed
Response: As suggested, we better discussed the differentiation of Th17 cells into Treg cells (new paragraph 2.2, lines 177-183)
5. Line 173, seems better to say “fully” elucidated as the authors go on to discuss evidence for IL-22 involvement in MS
Response: The suggested change has been done (new paragraph 2.4, line 229)
6. Re therapies: daclizumab was pulled from the market worldwide in 2018!
Response: We apologize for the mistake. Data on both daclizumab and alemtuzumab have been update following the information provided by EMA (new paragraph 4, lines 354-367, references 204 and 205)
Reviewer 2 Report
The manuscript „T helper cells: the modulators of inflammation in Multiple Sclerosis“ (Kunkl M. et al.) is well written review, which describes the main clinical forms of multiple sclerosis and the principal pathogenitic mechanisms that contribute to the development of inflammation and demyelination in CNS. The focus is made on the description of phenotypic and functional characteristics of encephalitogenic T helper cells (Th1, Th17, Th1-like Th17, Th9, and Th22 cells), as well as on the explanation of T cell targeting therapy that is currently used in the treatment of MS. Since the selected novel literature well documents the state of art in the field this review might be of a great interest both for the clinicians and for the scientists investigating the immunopathogenic mechanisms of MS and EAE.
Minor changes:
Line 3
Multiple Sclerosis should be multiple sclerosis
Lines 65-68
Th1 cells are a subset of CD4+ T cells identified in late 1980s [17,18] that have a key role in orchestrating efficient adaptive immune responses against intracellular pathogens by helping B lymphocytes to produce antibodies (Ab), cytotoxic T cells and by secreting specific cytokines in tissue, in particular IFN-g that activate macrophages to kill intracellular microbes [19].
Check the construction of the sentenceLine 103
Figure 1. Pathogenic Th cell subsets in MS. Self-reactive Th1, Th1-like Th17 and Th22 cell subsets
Consider to change into: Pathogenic Th cell subsets in MS. Self-reactive Th1, Th22 and Th1-like Th17 cell subsets to follow the order of subsets shown on the pictureAuthor Response
We thank the reviewer for suggestions and advices. To better clarify our responses, we numbered the reviewer’s comments and our responses are indicated below.
1. Line 3 Multiple Sclerosis should be multiple sclerosis:
Response: The suggested change has been done (new lines 3, 17 and 35)
2. Lines 65-68 Th1 cells are a subset of CD4+ T cells identified in late 1980s [17,18] that have a key role in orchestrating efficient adaptive immune responses against intracellular pathogens by helping B lymphocytes to produce antibodies (Ab), cytotoxic T cells and by secreting specific cytokines in tissue, in particular IFN-g that activate macrophages to kill intracellular microbes [19]. Check the construction of the sentence
Response: As suggested by reviewer, we modified the construction of the sentence (new paragraph 2.1, lines 79-81)
3. Line 103 Figure 1. Pathogenic Th cell subsets in MS. Self-reactive Th1, Th1-like Th17 and Th22 cell subsets. Consider to change into: Pathogenic Th cell subsets in MS. Self-reactive Th1, Th22 and Th1-like Th17 cell subsets to follow the order of subsets shown on the picture
Response: We modified the legend according to the reviewer suggestions and we also added more mechanistic insights in the Figure, as suggested by reviewer 4 (new Figure 1 and legend)
Reviewer 3 Report
The topic of the review is timely, the paper is well written.
Some points need to be clarified.
The authors describe that daclizumab was approved for the treatment of MS.
Further, the authors describe: “Due to several reported side effects, including high rate of infections, liver failure and autoimmune hepatitis [181,182], therapy with alemtuzumab or daclizumab has been approved for RRMS patients with high active disease who did not respond to first-line therapies [135].”
The description is wrong for both. Alemtuzumab might be used as first line therapy in high active patients as well. Daclicumab was withdrawn from the market due to cases of severe encephalitis!!!
The authors describe T cell targeting therapies and discuss nearly all therapeutics for MS except for ocrelizumab. Recently, it was published in Cells that ocrelizumab exerts direct effects on T cells. Beside depletion of B cells, ocrelizumab depletes a subset of CD20CD3 positive cells as well (Cells. 2018 Dec 28;8(1). pii: E12. doi: 10.3390/cells8010012.). This information should be incorporated and discussed.
Author Response
We thank the reviewer for suggestions and advices. To better clarify our responses, we numbered the reviewer’s comments and our responses are indicated below.
1. The authors describe that daclizumab was approved for the treatment of MS. Further, the authors describe: “Due to several reported side effects, including high rate of infections, liver failure and autoimmune hepatitis [181,182], therapy with alemtuzumab or daclizumab has been approved for RRMS patients with high active disease who did not respond to first-line therapies [135].” The description is wrong for both. Alemtuzumab might be used as first line therapy in high active patients as well. Daclicumab was withdrawn from the market due to cases of severe encephalitis!!!
Response: We apologize for the mistake. Data on both daclizumab and alemtuzumab have been update following the information provided by EMA (new paragraph 4, lines 354-367, references 204 and 205)
2. The authors describe T cell targeting therapies and discuss nearly all therapeutics for MS except for ocrelizumab. Recently, it was published in Cells that ocrelizumab exerts direct effects on T cells. Beside depletion of B cells, ocrelizumab depletes a subset of CD20CD3 positive cells as well (Cells. 2018 Dec 28;8(1). pii: E12. doi: 10.3390/cells8010012.). This information should be incorporated and discussed.
Response: As suggested, we discussed the direct effects of ocrelizumab on CD20+CD3+ T cells (new paragraph 4, lines 367-372)
Reviewer 4 Report
The manuscript submitted by Kunkl et al., covers the most relevant findings previously published in the context of CNS inflammatory demyelination of EAE and MS and T helper cells. The authors differentiate their review by subsets (Th1, Th1-like Th17, Th17, Th22 and Th9 cells), and finalize with a section specific for therapeutic strategies that target T helper cells. The authors included highly-cited previous work in their work, that is well written.
The authors focus their review on the immunopathogenesis associated with T helper cells. It would be interesting if the authors could incorporate a section focused on other T helper cell subsets associated with natural protection against neuroinflammation. Although there are references the Treg, it would be relevant to add a section for those cell populations that play a role controlling inflammation, such as IL-10 and TGF-b-producing Tregs, Foxp3-negative TGF-b producing Th3 cells, or Foxp3-negative IL-10-producing T cells.
It would be interesting to discuss the biological aspects by which the differentiation of these subsets occur in the context of CNS inflammation. perhaps the figure provided (Fig. 1) can be modified to add more mechanistic insights since, as shown, provides little relevant information.
Author Response
We thank the reviewer for suggestions and advices. To better clarify our responses, we numbered the reviewer’s comments and our responses are indicated below.
1. Although there are references the Treg, it would be relevant to add a section for those cell populations that play a role controlling inflammation, such as IL-10 and TGF-b-producing Tregs, Foxp3-negative TGF-b producing Th3 cells, or Foxp3-negative IL-10-producing T cells.
Response: We added a section for Treg, as suggested by reviewer (new paragraph 3). However, since the role of Treg in autoimmunity and multiple sclerosis has been finely and extensively reviewed by other groups, we just briefly described this population and cited the most recent and comprehensive reviews on the topic.
2. It would be interesting to discuss the biological aspects by which the differentiation of these subsets occur in the context of CNS inflammation. Perhaps the figure provided (Fig. 1) can be modified to add more mechanistic insights since, as shown, provides little relevant information.
Response: As suggested, we added more information about the biological effects of the distinct Th cell subsets in CNS, in particular Th1 and Th17 cells, in the CNS and modified Figure 1 accordingly (new Figure 1, new paragraph 2.1, lines 92-100, new paragraph 2.2, lines 132-149, new paragraph 2.3, lines 219-220)
Round 2
Reviewer 1 Report
My comments have been addressed.
The authors should validate minor typos: for example, line 201, the "t" in "RORgt" is in greek font.